# MPCFORMER: FAST, PERFORMANT AND PRIVATE TRANSFORMER INFERENCE WITH MPC

**Dacheng Li**[*c], **Rulin Shao**[*c], **Hongyi Wang**[*c], **Han Guo**[c], **Eric Xing**[mpc], **Hao Zhang**[b]

[c] Carnegie Mellon University   [m] Mohamed bin Zayed University of Artificial Intelligence
[p] Petuum Inc.  [b] University of California, Berkeley

## ABSTRACT

Enabling private inference is crucial for many cloud inference services that are based on Transformer models. However, existing private inference solutions can increase the inference latency by more than $60\times$ or significantly compromise the inference quality. In this paper, we design the framework MPCFORMER as a practical solution, using Secure Multi-Party Computation (MPC) and Knowledge Distillation (KD). Through extensive evaluations, we show that MPCFORMER significantly speeds up Transformer inference in MPC settings while achieving similar ML performance to the input model. On the IMDb dataset, it achieves similar performance to BERT$_{\text{BASE}}$, while being $5.3\times$ faster. On the GLUE benchmark, it achieves 97% performance of BERT$_{\text{BASE}}$ with a $2.2\times$ speedup. MPC-FORMER remains effective with different trained Transformer weights such as ROBERTA$_{\text{BASE}}$ and larger models including BERT$_{\text{Large}}$. Code is available at `https://github.com/MccRee177/MPCFormer`.

## 1 INTRODUCTION

Pre-trained Transformer models can be easily fine-tuned on various downstream tasks with high performance and have been widely developed as model inference services (Bommasani et al., 2021; Feng et al., 2020; Yang et al., 2019b; Clark et al., 2020). However, these model inference services can pose privacy concerns. For instance, GitHub Copilot, a code-generating engine adapted from pre-trained GPT weights, requires either users to reveal their code prompts to the service provider, or the service provider to release the Copilot's trained weights, which are business proprietary, to users (Chen et al., 2021; Brown et al., 2020).

Secure Multi-Party Computation (MPC) offers a promising solution by keeping data and model weights private during inference (Evans et al., 2018). However, the vanilla Transformer inference in MPC is unacceptably slow. For instance, BERT$_{\text{BASE}}$ inference takes <1 second without MPC, but $\sim$60 seconds with MPC (Figure 2). An intuitive way to accelerate MPC inference replaces computational operations with their faster approximations and retrains the approximated model, which has been adopted on convolutional neural networks (CNNs) (Chou et al., 2018). Unfortunately, adapting this solution to Transformers drastically decreases the model's performance (§ 5).

In this paper, we take the first step to pursue *privacy-preserving Transformer model inference in MPC*, while remaining *fast* and *performant*. We take inspiration from the approximation approach[1] and attribute the performance degradation to two challenges. First, many MPC-friendly approximations toughen model training. For example, quadratic functions cause the gradient explosion problem in deep neural networks (Mishra et al., 2020). Second, downstream datasets used for Transformer fine-tuning usually contain insufficient data to retrain an approximated Transformer with common task objectives (Zhang & Sabuncu, 2018; Hinton et al., 2012).

To address these two challenges, we resort to the *knowledge distillation* (KD) framework. KD can ease the model training by matching intermediate representations between the teacher and the student model (Romero et al., 2014); this intermediate supervision can alleviate the gradient explosion

---

[*]Authors contributed equally.

[1]We will use the term MPC-friendly approximations.

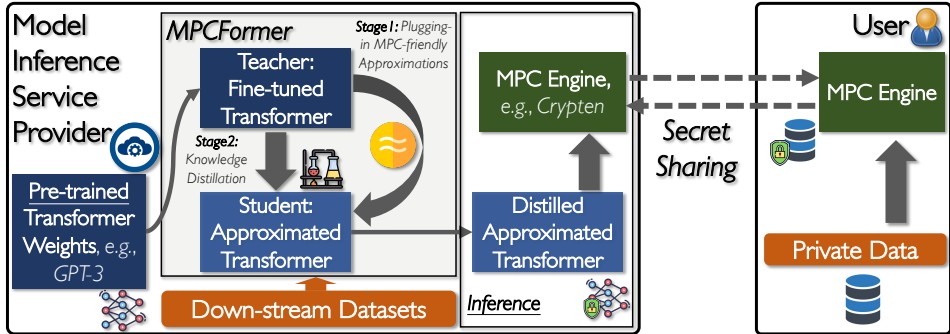

Figure 1: An illustration of our proposed MPCFORMER framework. MPCFORMER takes a trained (or finetuned) Transformer model and adopts given MPC-friendly approximations, then uses KD on the downstream datasets to construct high-quality models. During inference time, MPCFORMER leverages an MPC engine to attain private model inference. For ease of illustration, we only show the service provider and the user. MPC Systems such as CrypTen (Knott et al., 2021) may also involve a trusted third party (TTP) to help with the joint computation.

problem (Lee et al., 2015). At the same time, the KD objective is *data-efficient* and allows training an approximated Transformer on small downstream datasets (Touvron et al., 2021).

**Our approach and contributions.**    In this paper, we build MPCFORMER, an *easy-to-adopt* framework for privacy-preserving Transformer inference. MPCFORMER takes in an MPC-friendly approximation and a trained Transformer. It returns a Transformer with low inference latency in MPC and high ML performance simultaneously. To do so, MPCFORMER first replaces bottleneck functions in the input Transformer model with the given MPC-friendly approximations. The resulting approximated Transformer model has a faster inference speed in MPC. Next, it applies knowledge distillation to train the approximated Transformer with high performance, using teacher guidance from the original input Transformer. Finally, the model provider can use the distilled approximated Transformer on top of an MPC engine, *e.g.,* CrypTen, for private model inference service. The overall workflow of MPCFORMER is shown in Figure 1.

We implement MPCFORMER on an MPC system (Knott et al., 2021), with various MPC-friendly approximations. In the process, we also design a new and faster MPC-friendly approximation to the Softmax function. We extensively evaluate our implementation with various Transformer models. On the IMDb benchmark, MPCFORMER achieves similar ML performance to $\text{BERT}_{\text{BASE}}$ with a $5.3\times$ speedup. It achieves similar ML performance to $\text{BERT}_{\text{LARGE}}$ with a $5.9\times$ speedup. On the GLUE benchmark, it achieves 97% performance of $\text{BERT}_{\text{BASE}}$ with a $2.2\times$ speedup. MPCFORMER is also effective when given different trained Transformer models, *e.g.,* $\text{RoBERTa}_{\text{BASE}}$.

## 2    BACKGROUND

In this section, we first describe the Transformer model. Then we describe how functions in Transformer models can be implemented in MPC, and analyze performance bottlenecks.

### 2.1    TRANSFORMER MODELS

An $n$-layer Transformer model consists of three components: (1) The embedding layer. (2) A stack of $n$ Transformer layers. (3) The prediction layer. The embedding layer maps a token (e.g. a word or an image patch) to a hidden representation (Devlin et al., 2018; Dosovitskiy et al., 2020). One Transformer layer consists of an attention module and several matrix multiplication modules (Bahdanau et al., 2014). The prediction layer maps the output of the last Transformer layer to a task output (*e.g.,* a probability distribution for a classification task). A partial illustration of a Transformer model can be found in Figure 3.

### 2.2    TRANSFORMER MODELS IN MPC

The Transformer model inference process can be formulated as a 2-Parties Computation (2PC). In 2PC, the user party inputs the data, and the model provider party inputs the Transformer model.

They jointly compute an inference result. Throughout the entire inference process, 2PC guarantees both parties only know information about their own inputs and the result (Yang et al., 2019a).

We describe the **secret sharing** scheme as a mean to preserve privacy during the inference process (Damgård et al., 2012; Goldreich et al., 2019). Assuming that the user provides a number $x$ as its input, the secret sharing scheme splits $x$ into two numbers, $x_1$ and $x_2$. It then lets the user party hold $x_1$ and distributes $x_2$ to the model provider party. There are two properties of $x_1$ and $x_2$. First, either $x_1$ or $x_2$ alone contains no information about $x$. This property allows the user to hide the actual value $x$ from the model provider. Second, together they reconstruct $x$. For instance, $x_1$ and $x_2$ add up to $x$: $x = x_1 + x_2$. The second property allows joint computation.

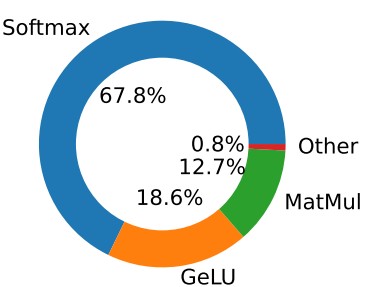

Figure 2: BERT$_{\text{BASE}}$ model (12-layers, 512 tokens) run-time breakdown evaluated on an MPC system. The overall runtime with MPC is 59.0 seconds, but $< 1$ second without MPC.

Table 1: The communication statistics explain the left figure. For instance, 50.3GB communication in Softmax functions takes 34.1 seconds. In particular, the run-time in MPC is dominated by the **communication** as opposed to computation. Overall, the communication takes 46.4 seconds, which is **79%** of the whole inference process.

| *Routines* | Comm. (rounds) | |
|---|---|---|
| Addition | 0 | |
| Multiplication | 1 | |
| Comparison | 7 | |
| *functions* | Comm. (volume) | Time (s) |
| MatMul | 3.5GB | 2.5 |
| GeLU | 14.8GB | 9.6 |
| Softmax | 50.3GB | 34.1 |

We take multiplication via Beaver triple as a joint computation example (Beaver, 1991). In multiplication, the user party provides $x$ and the model provider provides $y$, and they secret share $x$ and $y$. Thus, the user gets $x_1$ and $y_1$; the model provider gets $x_2$, and $y_2$. Beaver triple assumes a triple $c = ab$ has been generated[2]. The triple is also secret shared so that the user party gets $c_1, a_1, b_1$, and the model provider gets $c_2, a_2, b_2$. The user first computes $\epsilon_1 = x_1 - a_1$, $\delta_1 = y_1 - b_1$ locally. The model provider similarly computes $\epsilon_2 = x_2 - a_2$, $\delta_2 = y_2 - b_2$ locally. They communicate these four numbers and reconstruct $\epsilon = \epsilon_1 + \epsilon_2$, $\delta = \delta_1 + \delta_2$. the user then use these two values to compute $r_1 = c_1 + \epsilon b_1 + \delta a_1 + \delta\epsilon$. The model provider computes $r_2 = c_2 + \epsilon b_2 + \delta a_2$. At this point, the multiplication result $xy$ can be reconstructed by $xy = r_1 + r_2$.

There are two important observations in the multiplication example: (1) it does not leak information to the other party. For instance, the user does not send $x_1$ to the model party. Instead, it sends $\epsilon_1 = x_1 - a_1$ where $a_1$ is a random mask; (2) it requires one extra round of communication compared to the multiplication without MPC. This partially explains why vanilla Transformer models are slow in MPC. In particular, functions in Transformers (e.g., nonlinear activation) can be mainly implemented by three routines[3], i.e., addition, multiplication, and comparison. Any computational operations composed by these routines would result in extra complexity in MPC[4].

Empirically, we show these complexities by running BERT$_{\text{BASE}}$ (Figure 2) and reporting communication statistics in Table 1 with a secret-sharing-based MPC system (Knott et al., 2021). We observe that GeLU functions and Softmax functions in Transformer layers are the major sources of bottlenecks, which echoes findings in a concurrent study (Wang et al., 2022). $\text{GeLU}(x) = x \times \frac{1}{2} \left[ 1 + \text{erf}\left( \frac{x}{\sqrt{2}} \right) \right]$ is slow because the Gaussian Error function $\text{erf}(\cdot)$ is evaluated by a high order Taylor expansion, which requires many multiplication routines. $\text{Softmax}(\mathbf{x}_i) = \frac{\exp(\mathbf{x}_i)}{\sum_j \exp(\mathbf{x}_j)}$ is slow because (1) The exponential function is evaluated by several iterations of squaring, which requires many multiplication routines; (2) the maximum operation over $\mathbf{x}$ is required for numerical stability (Paszke et al., 2019), which requires comparison routines.

---

[2]For example, through oblivious transfer (Keller et al., 2016) or homomorphic encryption (Paillier, 1999).

[3]We only consider routines that take two secret share numbers for the ease of illustration.

[4]We provide more details on implementing routines and functions in MPC at A.1.

## 3    RELATED WORK

**MPC.**    Secure Multi-party Computation (MPC) enables joint computation between parties while keeping inputs private. The privacy feature and rich support of systems have made it suitable for Transformer inference (Mohassel & Zhang, 2017; Liu et al., 2017; Mohassel & Rindal, 2018; Riazi et al., 2018; Juvekar et al., 2018; Wagh et al., 2019; Mishra et al., 2020; Knott et al., 2021). In this paper, we do not aim to implement a new MPC system. Rather, we aim to develop an algorithmic solution to speed up Transformer inference that can be portable across many MPC systems.

**Transformer models.**    Transformer models have achieved great success in language understanding Yang et al. (2019b); Lan et al. (2019); Raffel et al. (2020); Clark et al. (2020), vision understanding Dosovitskiy et al. (2020); Liu et al. (2021); Radford et al. (2021), and beyond (Sharir et al., 2021). In particular, the two-stage training strategy for Transformer models has been shown to be effective in extensive settings and has become the domincated paradigm (Liu et al., 2019; Radford et al., 2018; Turc et al., 2019). In this training strategy, Transformer models are first pre-trained on a large dataset for general understanding, and then fine-tuned on a small downstream dataset to learn task-specific features. In this work, we consider this paradigm as the default setting, where we assume that model providers use pre-trained Transformer weights from elsewhere, and only have downstream data.

**MPC-friendly approximations.**    Existing research has developed MPC-friendly approximations to speed up CNN computation in MPC. Chou et al. (2018) develops an optimization framework that minimizes the approximation error of order 2 polynomial to ReLU: $\text{ReLU}(x) = 0.125 \times x^2 + 0.25 \times x + 0.5$. This introduces a significant accuracy drop because the quadratic activation causes the Gradient Descent (GD) algorithm to diverge. Mishra et al. (2020) alleviates this problem by using a set of carefully designed heuristics along with Neural Architecture Search (NAS). Mohassel & Zhang (2017) proposes an approximation to softmax by replacing exponential with ReLU functions. We do not focus on developing heuristics for a single pair of bottleneck functions and approximations. Rather, we focus on developing a general framework that can consistently output a performant Transformer model with various approximations.

**Knowledge Distillation (KD).**    KD transfers knowledge from the teacher model to the student model by matching their hidden representations (Hinton et al., 2006). Several research has designed effective objectives for Transformer models (Sanh et al., 2019; Jiao et al., 2019; Dosovitskiy et al., 2020) such as matching the attention matrices. In particular, Sanh et al. (2019) and Jiao et al. (2019) have a different goal than us and train on the pre-training dataset as well. However, we share the same assumption on the model providers' side — they only have downstream datasets.

## 4    METHOD

In this section, we present the MPCFORMER framework. MPCFORMER allows the model provider to convert its Transformer model to a faster and performant one for private inference service. In 4.1, we introduce the workflow of MPCFORMER (Figure 1), followed by the details of each step in 4.2.

### 4.1    HIGH-LEVEL WORKFLOW

In the inference service, a model provider holds a Transformer model $\mathcal{T}$, and the user holds data $X$. They reach to an agreement on an MPC system to perform private inference. In §2, we illustrate that using $\mathcal{T}$ to perform the inference in the MPC system is slow. Instead of using $\mathcal{T}$, the model provider can use MPCFORMER to generate a more suited one $\mathcal{S}$. $\mathcal{S}$ runs much faster than $\mathcal{T}$ in the MPC setting while having similar ML performance compared to $\mathcal{T}$.

To use MPCFORMER, the model provider needs to provide the trained Transformer model $\mathcal{T}$, the downstream dataset $\mathcal{D}$, and MPC-friendly approximations $\mathcal{A}$. These MPC-friendly approximations $\mathcal{A}$ need to be fast in MPC and will be used to replace bottleneck functions in $\mathcal{T}$. The workflow can be concisely described as:

$$\text{Convert:}\quad \mathcal{S} = \text{MPCFORMER}(\mathcal{T}, \mathcal{D}, \mathcal{A})$$
$$\text{Inference:}\quad y = \text{MPC}_{\mathcal{S}}(X) \tag{1}$$

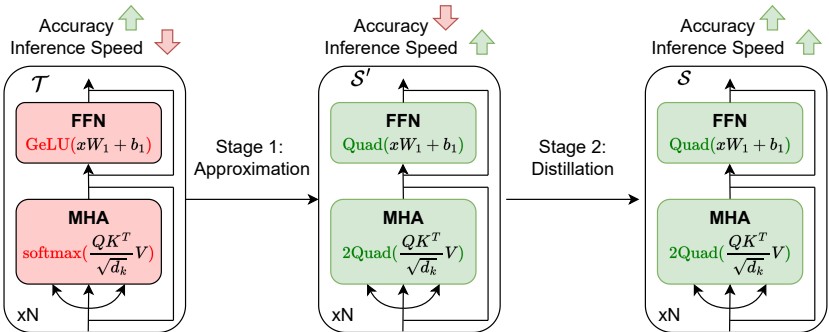

Figure 3: The overview of the MPCFORMER framework. The first stage uses MPC-friendly approximations and $\mathcal{T}$ to construct a faster Transformer architecture $\mathcal{S}'$. The second stage uses Knowledge Distillation on $\mathcal{S}'$ to learn a performant and fast Transformer model $\mathcal{S}$.

## 4.2 MPCFORMER

MPCFORMER is a two-stage framework as shown in Figure 3. The first stage leverages $\mathcal{A}$ and $\mathcal{T}$ to construct an MPC-friendly Transformer architecture $\mathcal{S}'$, which achieves *fast* inference in the MPC system. The second stage applies knowledge distillation (KD) to $\mathcal{S}'$ to learn the output model $\mathcal{S}$, which is fast in MPC *and* preserves the *high performance* of $\mathcal{T}$.

### 4.2.1 STAGE 1: APPROXIMATION

In the first stage, MPCFORMER replaces bottleneck functions in $\mathcal{T}$ with given $\mathcal{A}$ to construct a MPC-friendly Transformer architecture $\mathcal{S}'$ (Figure 3). Below we show how we construct $\mathcal{A}$ for our experiments *i.e.,* using the MPC system Knott et al. (2021).In §2, we identify the bottleneck to be GeLU and Softmax functions. We thus construct $\mathcal{A}$ for these two functions.

**Approximating GeLU.** Analysis in § 2 shows that multiplication in MPC requires extra communication. Thus, quadratics are the fastest nonlinear activation in MPC. Since GeLU and ReLU functions share similar function values, we simply take the quadratics designed for the ReLU function (§3) to approximate the GeLU function: $GeLU(x) \approx 0.125x^2 + 0.25x + 0.5$. We denote this approximation as "Quad".

**Approximating Softmax.** Prior works in CNNs have developed an MPC-friendly approximation to Softmax functions (Mohassel & Zhang, 2017):

$$\text{softmax}(x) \approx \text{ReLU}(x)/\sum \text{ReLU}(x) \tag{2}$$

We validate that this has a faster inference speed than the Softmax function in our setting (Figure 4). We denote this approximation as "2ReLU". However, this is not yet satisfactory. Analysis in §2 shows that evaluating the ReLU function requires heavy use of comparison routines, which is very expensive. Thus, we propose a more aggressive approximation for the Softmax by replacing the ReLU in Eq. 2 with a quadratic function:

$$\text{softmax}(x) \approx (x + c)^2/\sum (x + c)^2 \tag{3}$$

We denote this as "2Quad". Importantly, "2Quad" and Softmax functions differ a lot by numerical values, while prior works argue that similarity in numerical values is crucial to the model's performance (Chou et al., 2018). We are able to use this aggressive approximation because our next distillation stage is effective enough to bridge the performance gap. Figure 4 shows the comparison between the running time of the original GeLU and softmax function against their approximations. In particular, 2Quad has a much faster inference speed than 2ReLU.

### 4.2.2 STAGE 2: DISTILLATION

In the second stage, we use KD to make the fast approximated Transformer model $\mathcal{S}'$ performant. The benefits of KD are two-fold. First, it allows us to use more aggressive approximations such as "2Quad", which leads to higher speedups. Second, its data efficiency allows us to effectively learn a good $\mathcal{S}$ with the small downstream datasets. Concretely, we conduct layer-wise distillation

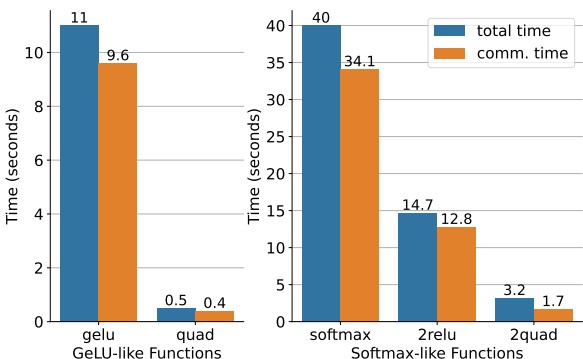

Figure 4: Running time comparison of different approximations in §4.2.1. Blue areas represent the total running time. Orange areas represent the communication time. MPC-friendly approximations greatly reduce both the communication time and the total time. In particular, our proposed "2Quad" is much faster than the original Softmax function, and the previous "2ReLU" approximation.

to transfer knowledge from the input model $\mathcal{T}$ to $\mathcal{S}'$ by matching representation at the following four positions: (1) the embedding layer, (2) the attention matrix in each Transformer layer, (3) the hidden states after each Transformer layer, and (4) the final prediction layer. These four positions have been shown to store meaningful information in previous works (Hinton et al., 2015; Jiao et al., 2019; Clark et al., 2019). We use the Mean Square Error (MSE) loss to match the representations between $\mathcal{T}$ and $\mathcal{S}$ for all positions. We follow the learning procedure of Jiao et al. (2019) to first distill the embedding and Transformer layers (including the attention matrix and the hidden states) and then distill the prediction layer.

**Student initialization.** An important component of knowledge distillation is the initialization of the student model (Sanh et al., 2019). Taking the advantage that $\mathcal{S}'$ and $\mathcal{T}$ share the same architecture, we initialize $\mathcal{S}'$ using weights in $\mathcal{T}$. We find that this outperforms random weight initialization, especially on smaller datasets (§5.3).

## 5 Experiments

We design the MPCFORMER framework to be compatible with many MPC-friendly approximations and trained Transformer models, so that model providers can conveniently plug in MPC-friendly approximations based on their MPC systems. Thus, we evaluate MPCFORMER with different MPC-friendly approximations under (1) Different datasets (§ 5.1), and (2) Different models (especially larger models) (§5.2). In the ablation study, we study (1) the effect of student initialization and, (2) the effect of the number of training examples in the distillation stage.

**Experimental setup.** We use two P3.2x AWS instances to simulate the inference service scenarios (one P3.2x for the model provider, and one for the user). Each instance is equipped with one Tesla V100 GPU, and 10GbE Ethernet bandwidth. We place instances in the same placement group to guarantee a 10GbE bandwidth in AWS. Time breakdown is measured with CrypTen, which implements secret sharing with semi-honest adversaries assumption (Section §2) (Knott et al., 2021). We train and evaluate models based on HuggingFace (Wolf et al., 2019). In particular, we find that the implementation of 2Quad requires careful use of HuggingFace source code (§A.2).

**Baselines.** We identify three important properties during Transformer inference in § 1: speed, performance, and privacy. In our workflow, privacy has been guaranteed by using MPC systems. Thus, we evaluate MPCFORMER by the other two aspects: speed and performance. Concretely, $\mathcal{S}$ shall run faster than $\mathcal{T}$ while matching the performance of $\mathcal{T}$. Since there is a limited amount of work on Transformer inference in MPC, we seek a baseline from CNN literature. In particular, we choose the training strategy in (Chou et al., 2018) and denote it as MPCFORMER$_{\mathrm{w/o\{d\}}}$. MPCFORMER$_{\mathrm{w/o\{d\}}}$ also constructs the approximated model $\mathcal{S}'$ but trains $\mathcal{S}'$ on $\mathcal{D}$ with the task-specific objective, *i.e.*, without distillation. We note that $\mathcal{S}'$ is initialized with weights in $\mathcal{T}$, *i.e.*, with different functions, whose effect has not been studied. We thus propose a second baseline MPC-

FORMER$_{w/o\{p,d\}}$, which trains $\mathcal{S}'$ on $\mathcal{D}$ without distillation, and random weight initialization. Below we compare the performance of MPCFORMER with MPCFORMER$_{w/o\{p,d\}}$ and MPCFORMER$_{w/o\{d\}}$ with the same speedups.

We denote the output model of our framework with BERT$_{BASE}$, Roberta-base, and BERT$_{LARGE}$ as MPCBert-B, MPCRoberta-B, and MPCBert-L for short.

## 5.1 Comparison with Baselines on Different Benchmarks

**Settings.** In this section, we evaluate our MPCFormer framework with different approximations and compare it with baselines on the IMDb dataset and the GLUE benchmark (Maas et al., 2011; Wang et al., 2018). For all experiments in this section, we use BERT$_{BASE}$ as the base model. According to the dataset statistics, we use a sequence length of 512 for the IMDb dataset and a sequence length of 128 for GLUE datasets. We note that a longer sequence length generally reflects a higher speedup because the Softmax functions can be sped up more. Baselines are trained with learning rates tuned from 1e-6, 5e-6, 1e-5, and 1e-4, the number of epochs from 10, 30, and 100, the batch size 32 for IMDB, batch sizes 64 and 256 for GLUE. MPCBert-B is trained with learning rate 5e-5 for embedding and Transformer layer distillation, and 1e-5 for prediction layer distillation. Further details on hyper-parameters tuning can be found in A.4.

Table 2: Performance and speedup on the IMDB dataset and a part of the GLUE benchmark (QNLI, CoLA and RTE) with different approximations and BERT$_{BASE}$ as the backbone. The input model $\mathcal{T}$ is denoted with "*". "p" stands for using weights in $\mathcal{T}$ as initialization, "d" stands for applying knowledge distillation with $\mathcal{T}$ as the teacher.

| Method | Approximation | IMDb | | GLUE | |
|---|---|---|---|---|---|
| | | Speedup | Accuracy | Speedup | Avg. Score |
| Bert-B* | GeLU+Softmax | 1× | 94.1 | 1× | 73.1 |
| MPCBert-B$_{w/o\{p,d\}}$ | | | 87.5 | | 40.8 |
| MPCBert-B$_{w/o\{d\}}$ | Quad+Softmax | 1.24× | 87.5 | 1.13× | 43.0 |
| **MPCBert-B** | | | **94.0** | | **72.6** |
| MPCBert-B$_{w/o\{p,d\}}$ | | | 86.1 | | 39.6 |
| MPCBert-B$_{w/o\{d\}}$ | GeLU+2ReLU | 1.76× | 93.8 | 1.40× | 71.8 |
| **MPCBert-B** | | | **94.0** | | **72.0** |
| MPCBert-B$_{w/o\{p,d\}}$ | | | 85.8 | | 43.5 |
| MPCBert-B$_{w/o\{d\}}$ | Quad+2ReLU | 2.61× | 86.8 | 1.93× | 48.2 |
| **MPCBert-B** | | | **94.0** | | **69.8** |
| MPCBert-B$_{w/o\{p,d\}}$ | | | 87.3 | | 39.6 |
| MPCBert-B$_{w/o\{d\}}$ | GeLU+2Quad | 2.65× | 90.6 | 1.55 × | 69.7 |
| **MPCBert-B** | | | **94.0** | | **71.0** |
| MPCBert-B$_{w/o\{p,d\}}$ | | | 87.8 | | 40.7 |
| MPCBert-B$_{w/o\{d\}}$ | **Quad+2Quad** | **5.26×** | 87.3 | **2.20×** | 40.8 |
| **MPCBert-B** | | | **93.9** | | **68.4** |

Table 3: Performance and speedup on the IMDB dataset and a part of the GLUE benchmark (QNLI, CoLA and RTE) with different approximations and BERT$_{BASE}$ as the backbone. The input model $\mathcal{T}$ is denoted with "*". "p" stands for using weights in $\mathcal{T}$ as initialization, "d" stands for applying knowledge distillation with $\mathcal{T}$ as the teacher.

| Method | IMDb | | GLUE | |
|---|---|---|---|---|
| | Speedup | Accuracy | Speedup | Avg. Score |
| Solution 1 | 1× | 94.1 | 1× | 73.1 |
| Solution 2 | **5.26×** | 87.8 | **2.20×** | 40.7 |
| **MPCFormer** | | **93.9** | | **68.4** |

We show the accuracy and speedup on the IMDb dataset in Table 3. MPCBert-B achieves 5.26× speedup with "Quad+2Quad" approximation with almost no accuracy drop. In addition, we note that this holds for not only the fastest "Quad+2Quad" approximation but other approximations rang-

Table 4: Performance on Glue benchmark with BERT$_{BASE}$ as the backbone. F1 score is reported for QQP and MRPC. Average Pearson and Spearman correlation is reported for STS-B. Matthews correlation is reported for CoLA. Accuracy is reported for other datasets.

| Method | Approx. | MNLI 393k | QQP 363k | QNLI 108k | SST-2 67k | CoLA 8.5k | STS-B 5.7k | MRPC 3.5k | RTE 2.5k | Avg. | Speed -up |
|--------|---------|------|-----|------|------|------|-------|------|-----|------|-------|
| Bert-B$^*$(GeLU+Softmax) | | 84.7/85.0 | 88.1 | 91.7 | 93.1 | 57.8 | 89.1 | 90.3 | 69.7 | 82.8 | 1× |
| MPCBert-B$_{w/o \{p,d\}}$ | **Quad** | 62.1/61.3 | 74.6 | 61.8 | 80.7 | 13.4 | 23.1 | 81.2 | 55.2 | 56.6 | |
| MPCBert-B$_{w/o \{d\}}$ | **+** | 73.1/72.5 | 82.9 | 75.5 | 83.4 | 16.4 | 41.3 | 81.2 | 52.7 | 63.3 | **1.93×** |
| **MPCBert-B** | **2ReLU** | **85.0/85.3** | **87.8** | **91.2** | **92.0** | **54.0** | **85.7** | **88.9** | **64.3** | **81.1** | |
| MPCBert-B$_{w/o \{p,d\}}$ | **Quad** | 63.5/62.4 | 78.6 | 59.8 | 81.1 | 9.9 | 19.5 | 81.4 | 52.7 | 55.7 | |
| MPCBert-B$_{w/o \{d\}}$ | **+** | 70.6/70.5 | 83.4 | 69.8 | 83.3 | 0 | 36.1 | 81.2 | 52.7 | 60.9 | **2.2×** |
| **MPCBert-B** | **2Quad** | 84.9/85.1 | 88.1 | 90.6 | 92.0 | 52.6 | 80.3 | 88.7 | 64.9 | 80.3 | |

ing from 1.24× to 2.65 × speedups. Baselines have inferior performance for all approximations. For example, both baselines have at least an accuracy drop of 6.8% with 5.26 × speedup. Interestingly, MPCBert-B$_{w/o \{d\}}$ has moderate accuracy drop for "GeLU+2ReLU" approximation with 1.76× speedup. However, it does not preserve accuracy with other approximations. In contrast, MPCFORMER consistently preserves accuracy with various kinds of approximation.

To validate the observation with more datasets, We evaluate MPCBert-B on the Glue benchmark (8 datasets) (Wang et al., 2018). As shown in Table 4, MPCBert-B achieves 1.93× speedup with 98% performance of BERT$_{BASE}$ on the GLUE benchmark, and 2.2× speedup with 97% performance of BERT$_{BASE}$. Both baselines introduce severe performance drop, i.e., 19.5 average score drop for MPCBert-B$_{w/o\{d\}}$ and 26.2 average score drop for MPCBert-B$_{w/o\{p,d\}}$ with 1.93× speedup. Interestingly, we observe that the baseline MPCBert-B$_{w/o\{d\}}$ consistently outperforms MPCBert-B$_{w/o\{p,d\}}$. This indicates that using weights in $\mathcal{T}$ as initialization benefits the training of $\mathcal{S}'$ with a task-specific objective, using a 12-layer Transformer backbone. Additionally, we evaluate MPCFORMER with more approximations using a subset of the GLUE benchmark. Results are shown in the right part of Table 3. We observe similar patterns as in the IMDb dataset. The baseline MPCBert-B$_{w/o\{d\}}$ performs well in "GeLU+2Quad' and "GeLU+2ReLU" approximations, but MPCFORMER achieves high ML performance consistently under all approximations.

## 5.2 MORE COMPARISONS WITH DIFFERENT MODELS

We evaluate MPCFORMER with trained Transformer models other than BERT$_{BASE}$, *i.e.,* ROBERTA$_{BASE}$ model (12 layers) (Liu et al., 2019), and in particular a larger BERT$_{LARGE}$ model (24 layers). Results are shown in Table 5. MPCRoberta-B preserves 98% average score of the input model ROBERTA$_{BASE}$, and outperforms baselines by large margins. Comparing the performance of baselines, we again observe that initialization with weights in $\mathcal{T}$ helps training with $\mathcal{S}'$, *i.e.,* MPCRoberta-B$_{w/o \{d\}}$ performs better than MPCRoberta-B$_{w/o \{p,d\}}$.

Table 5: The performance on a subset of Glue benchmark with Roberta-base backbone (denoted as "MPCRoberta-B"). MPCRoberta-B and baselines use "Quad+2Quad" approximations with 2.1 × speedup. MPCBert-L and baselines use "Quad+2ReLU" approximations with 2.0 × speedup.

| | MNLI-m | MNLI-mm | QNLI | RTE | Avg | Speedup |
|--|--------|---------|------|-----|-----|---------|
| Roberta-B$^*$ | 87.4 | 87.0 | 92.6 | 76.5 | 85.8 | 1× |
| MPCRoberta-B$_{w/o \{p,d\}}$ | 58.0 | 58.1 | 69.0 | 52.7 | 59.5 | |
| MPCRoberta-B$_{w/o \{d\}}$ | 73.1 | 72.7 | 81.6 | 52.7 | 70.0 | **2.1×** |
| **MPCRoberta-B** | **86.5** | **86.9** | **92.2** | **72.2** | **84.5** | |
| Bert-L$^*$ | 86.7 | 86.6 | 92.7 | 75.1 | 85.3 | 1× |
| MPCBert-L$_{w/o \{p,d\}}$ | 62.3 | 62.3 | 60.0 | 52.7 | 59.3 | |
| MPCBert-L$_{w/o \{d\}}$ | 35.4 | 35.2 | 50.5 | 52.7 | 43.5 | **2.0×** |
| **MPCBert-L** | **86.5** | **86.7** | **92.8** | **72.2** | **84.6** | |

To show our method can scale to different model sizes, we evaluate MPCFORMER with a larger model BERT$_{LARGE}$ (24-layer). On the IMDb dataset, BERT$_{BASE}$ achieves 95.0% accu-

racy. MPCBert-L achieves 5.9× speedup with 94.5% accuracy, while baselines MPCBert-L$_{\text{w/o }\{p,d\}}$ achieves 87.1% accuracy, and MPCBert-L$_{\text{w/o }\{d\}}$ achieves 50.0% accuracy. We further select four datasets from the GLUE benchmark, where Bert-L* noticeably outperforms Bert-B*(MNLI, QNLI, CoLA, and RTE). Compared with Bert-B* in table 4, Bert-L* increases the average score from 82.8 to 85.3. MPCFORMER increases the average score from 81.5 to 84.6. This indicates that MPCFORMER can scale with the size of $\mathcal{T}$. On the other hand, baselines do not scale with the input model: MPCBert-L$_{\text{w/o }\{p,d\}}$ decreases the average score from 60.1 to 59.3; MPCBert-L$_{\text{w/o }\{d\}}$ decreases the average score from 68.5 to 43.5. In particular, we observe that initializing $\mathcal{S}'$ with weights in $\mathcal{T}$ without distillation harms performance when the model is larger.

## 5.3 Ablation study

The first question we study is the effect of different student initialization. This is of interest as we do not have a general knowledge of whether initializing with weights in $\mathcal{T}$ will still benefit the training of $\mathcal{S}'$ after aggressive approximations. We design an experiment with random initialization (denoted as MPCBert-B$_r$ in the table). We train MPCBert-B$_r$ with 10× more epochs than MPCBert-B$_r$ and confirm that its distillation objective has converged. We observe that on larger datasets(QNLI and SST-2), the gap between using different initialization is small, but on smaller datasets(STS-B, MRPC, and RTE), initializing with the weights in $\mathcal{T}$ is better.

The second question we study is the effect of the number of training examples in the distillation stage. We perform studies on two small (RTE, MRPC) and two medium (SST-2, QNLI) datasets in the GLUE benchmark (Figure 5). We find that roughly 5% of the small datasets and 2% of the medium datasets provides enough data to learn a good $\mathcal{S}$. This shows that KD in our setting is efficient enough to learn a good $\mathcal{S}$ in downstream tasks (using GLUE datasets as representatives).

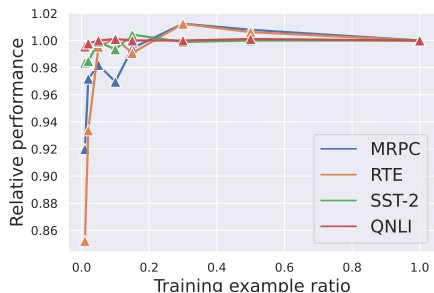

Figure 5: Ratio of training examples versus performance (normalized by performance with ratio = 1.0) on MRPC, RTE, QNLI, and SST-2.

Table 6: Student model initialized with weights in $\mathcal{T}$ versus with random weights. Result for MPCBert-B$_r$ is tuned with embedding and Transformer layer distillation learning rates from 5e-5 and 3e-5. Results for both are obtained with the "Quad+2Quad" approximation.

|  | QNLI | SST-2 | STS-B | MRPC | RTE | Avg |
|---|---|---|---|---|---|---|
| MPCBert-B | **90.6** | **92.0** | **80.8** | **88.7** | **64.9** | **84.0** |
| MPCBert-B$_r$ | **90.6** | 90.0 | 60.0 | 81.2 | 58.5 | 76.1 |

## 5.4 Limitation and future direction

We recognize two limitations in our paper. First, our speedups and performance are tested on a single MPC system. We leave theoretical analysis or empirical study on more MPC systems as future work. Second, in our design, $\mathcal{T}$ and $\mathcal{S}$ only differ by functions, *i.e.,* they have the same model size. We leave extension to a smaller student model as future work.

## 6 Conclusion

In this paper, we propose a framework to achieve fast and performant private Transformer model inference with MPC. Evaluations show that it is compatible with various MPC-friendly approximations and trained Transformer models. We suggest two directions of interest: (1) Theoretical or empirical analysis on more MPC systems, and (2) extension on the problem formulation to allow a smaller size of $\mathcal{S}$.

## Acknowledgement

The authors thank Qirong Ho for allocating the computing resources. This research was supported by NSF IIS1563887, NSF CCF1629559, NSF IIS1617583, NGA HM04762010002, NSF IIS1955532, NSF CNS2008248, NSF IIS2123952, and NSF BCS2040381.

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

# A APPENDIX

## A.1 A CONCRETE SYSTEM IMPLEMENTATION OF MPC: CRYPTEN

In this section, we provide how a concrete MPC system (CrypTen) implements routines and functions for Transformer models in detail (Knott et al., 2021). We provide a portion of details here to help describe the complexity of Transformer inference in MPC. A more complete system overview and privacy proof are available in the CrypTen paper.

**Threat model.** CrypTen follows Evans et al. (2018) to assume that parties are semi-honest. Under this assumption, parties are *honest* that they will follow the system protocols. However, each party is also *curious* (i.e., semi-honest), meaning it will try to infer the information about others' data based on the values it receives.

**Secret shares** CrypTen uses secret shares to implement private computation. A floating point value $x_f$ is first scaled to an integer $x^5$, then secretly shared with both parties. Secret shares are of type *arithmetic* or *binary*. The arithmetic secret shares $[x] = \{[x]_1, [x]_2\}$ is a set of two numbers, where the first party holds $[x]_1$, and the second holds $[x]_2$. They are constructed with a pair of zero-sum random maskings (Cramer et al., 2005) so that $[x]_1 + [x]_2 = x$. Binary shares $\langle x \rangle$ are formed by arithmetic secret shares of bits in x, so that the bitwise xor $\langle x \rangle_1 \oplus \langle x \rangle_1 = x$.

**Routines and functions** In arithmetic secret shares, to privately evaluate addition $[x] + [y]$, both parties simply compute: $[x]_i + [y]_i$ individually. Multiplication $[x][y]$ is evaluated by using a Beaver triple generated off-line $([c], [a], [b])$ where $c = ab$ (Beaver, 1991). Both parties compute and reveal intermediate values $[\epsilon] = [x] - [a]$ and $[\delta] = [y] - [b]$. The final result is computed by $[x][y] = [c] + \epsilon[b] + [a]\delta + \epsilon\delta$. Linear functions such as matrix multiplication can be evaluated by using additions and multiplications. Non-linear functions are evaluated by numerical approximations using additions and multiplications, such as Taylor expansion.

Comparison requires conversions between arithmetic and binary secret shares. Conversion from $[x]$ to $\langle x \rangle$ first creates binary secret shares $\langle [x]_i \rangle$ from arithmetic secret share $[x]_i$ $(i = 0, 1)$, then computes $\langle x \rangle = \langle [x]_1 \rangle + \langle [x]_2 \rangle$ using an adder circuit. Conversion from $\langle x \rangle$ to $[x]$ is done by: $[x] = \sum_{b=1}^{B} 2^b [\langle x \rangle^{(b)}]$, where $\langle x \rangle^{(b)}$ is the $b^{th}$ bit of $\langle x \rangle$. The comparison function $[z < 0]$ is then evaluated by: (1) convert $[z]$ to $\langle z \rangle$ (2) compute the sign bit $\langle b \rangle = \langle z \rangle >> (L - 1)$. (3) Convert $\langle b \rangle$ to $[b]$.

We study on the standard setting where each tensor is represented in 64 bits (i.e. L=64). Each multiplication requires one round of communication for revealing the intermediate values $\epsilon, \delta$. Each conversion from $[x]$ to $\langle x \rangle$ requires $\log_2 L = 6$ rounds of communications for the adder circuit; each conversion from $\langle x \rangle$ to $[x]$ requires one round for generating $[\langle x \rangle^{(b)}]$. Thus, each comparison requires 7 rounds of communication. Each $\max(\cdot)$ between N elements requires $\mathcal{O}(\log_2(N))$ rounds of communications, assuming a tree-reduction algorithm.

We provide a simple addition example here. The scaling factor and ring size $\mathcal{Q}$ are set to small for ease of understanding.

Table 7: Example of private addition computation. Suppose m is the actual message, then each party holds a share of m such that: $[m]_1 + [m]_2 = m$.

| Action | Party 1 | Party 2 | Note |
|---|---|---|---|
| Declare x | $x = 1$ | $x = $ random | x provided by party 1 |
| Generate a secret-sharing mask for x | $[z_x]_1 = -4$ | $[z_x]_2 = 4$ | sum to 0 |
| secret share x | $[x]_1 = x + [z_x]_1 = -3$ | $[x]_2 = [z_x]_2 = 4$ | sum to $x_1$ |
| Declare y | $y = $ random | $y = 2$ | y provided by party 2 |
| Generate a secret-sharing mask for y | $[z_y]_1 = 50$ | $[z_y]_2 = -50$ | sum to 0 |
| secret share y | $[y]_1 = [z_y]_1 = 50$ | $[y]_2 = y + [z_y]_2 = -48$ | sum to $y_2$ |
| Compute $x + y$ | $[x + y]_i = [x]_1 + [y]_1 = 47$ | $[x + y]_2 = [x]_2 + [y]_2 = -44$ | |
| Reveal $x + y$ | $x + y = [x + y]_1 + [x + y]_2 = 3$ | $x + y = [x + y]_1 + [x + y]_2 = 3$ | both get correct results |

---

[5] $x \in \mathcal{Z}/\mathcal{QZ}$ is required for privacy protocols, where $\mathcal{Z}/\mathcal{QZ}$ is a ring with $\mathcal{Q}$ elements.

In addition, we provide a more complete breakdown in terms of communication and computer load for Figure 2 for a holistic view of execution pattern in the MPC setting.

Table 8: Functions computation versus communication breakdown (Unit: seconds).

| functions | Comm. Time | Comp. Time | Total Time |
|---|---|---|---|
| MatMul | 2.5 | 5.0 | 7.5 |
| GeLU | 9.6 | 1.4 | 11.0 |
| Softmax | 34.1 | 5.9 | 40.0 |
| Others | 0.3 | 0.2 | 0.5 |
| Total | 46.5 | 12.5 | 59.0 |

In particular, computation only takes 21% of the running time and the communication takes 79% of the running time. Consequently, the number of floating point operations (FLOP), a popular estimator for the running time in plain-text Transformer inference, is no longer accurate in the MPC setting. The MPC system we use in the paper uses All-reduce to implement intermediate communication, where both parties have the same communication load. And they have similar computation load (see the multiplication example, where both parties are computing the same function locally with their own secret shares). Thus, the time breakdown is similar for both parties. In the above table, we report the statistics from the model provider.

## A.2   2QUAD IMPLEMENTATION DETAILS

We note that, implementing "2Quad" to replace the softmax requires attention to the effect brought by the masking operation. For example, the default implementation by Huggingface Wolf et al. (2019) would result in an exploding problem due to masking. Therefore, we would need to do a different version of the implementation of masking. We describe it in detail below.

The default attention implementation by Huggingface is

$$Attention(Q, K, V) = softmax(\frac{QK^T}{\sqrt{d_k}} + M_{\{0,-inf\}})V$$

$$= \frac{e^{\left(\frac{QK^T}{\sqrt{d_k}} + M_{\{0,-inf\}}\right)}}{\sum_{j=1}^{K} e^{\left(\frac{QK^T}{\sqrt{d_k}} + M_{\{0,-inf\}}\right)_j}} V.$$

If we directly replace the $e^x$ with $(x+c)^2$ as in 2Quad approximation, where $x = \frac{QK^T}{\sqrt{d_k}} + M_{\{0,-inf\}}$ will explode when being masked, causing a problem in the forward pass. To solve this problem, we could simply change the implementation of masking from "adding a zero or negative infinite number in the exponent" to "multiplying one or zero to the exponential function". That is,

$$Attention(Q, K, V) = \frac{e^{\left(\frac{QK^T}{\sqrt{d_k}}\right)} \odot M_{\{1,0\}}}{\sum_{j=1}^{K} e^{\left(\frac{QK^T}{\sqrt{d_k}}\right)_j} \odot M_{\{1,0\}}} V$$

$$\rightarrow \frac{\left(\frac{QK^T}{\sqrt{d_k}} + c\right)^2 \odot M_{\{1,0\}}}{\sum_{j=1}^{K} \left(\frac{QK^T}{\sqrt{d_k}} + c\right)_j^2 \odot M_{\{1,0\}}} V.$$

It's just a different implementation of the same masking purpose but avoids exploding at the masking positions.

In our experiments, we empirically tried $c = 5$ and it worked pretty well, indicating the choice of the constant $c$ could be flexible.

### A.3 ROBUSTNESS OF THE STUDENT MODEL

Some approximations may increase the local Lipschitz constants, which decreases the robustness. We applied some empirical text adversarial attacks to evaluate the adversarial robustness of the BERT-base model before and after approximations (Zeng et al., 2020). As shown in Table 9, the student model has a moderate increase in terms of attack success rate (ASR) over the three score-based attacks. But the student model has a lower ASR with the gradient-based attack HotFlip. Considering these results, the effect on robustness by the approximations are empirically moderate.

Table 9: Sanity accuracy (SA) and attack success rate (ASR) against various text attacks. The TextFooler, PWWS, and BERT-ATTACK are score-based attacks, and HotFlip is a gradient-based attack. For ASR, lower is better.

|  | SA | TextFooler | PWWS | BERT-Attack | HotFlip | Speedup |
|---|---|---|---|---|---|---|
| Bert-base (teacher) | **0.93** | **0.76** | **0.78** | **0.88** | 0.55 | 1.0x |
| Bert-base (student) | 0.92 | 0.78 | 0.82 | 0.91 | **0.51** | **2.2x** |

### A.4 HYPER-PARAMETER CHOICE

For baselines, We study the effect of hyper-parameters by running a grid search over the STS-B dataset [6], with learning rate from [1e-6, 5e-6, 1e-5, 5e-5, 1e-4, 5e-4], batch size from [256, 128, 64, 32, 16], epoch from [3, 10, 30, 50, 80, 100, 200]. We show the grid search results with $BERT_{BASE}$ in figure 6, 7, and a smaller grid search for $BERT_{Large}$ and $ROBERTA_{BASE}$ in Figure 8, 9. We empirically discover that the learning rates from 1e-6, 5e-6, 1e-5, batch size from 64 and 256, epoch from 10, 100 give good performance. To let baselines explore more hyper-parameters, we use learning rate from [1e-6, 5e-6, 1e-5, 1e-4], batch size from [64, 256], epochs from [10, 30, 100] for all Glue datasets. Since we use sequence length 512 for IMDB dataset, we use batch size 32 to fit into our 16GB Tesla V100 GPU. We also empirically discover that (1) MPCBert-B$_{w/o\{d\}}$ (best 0.43) can not scale up when the base model scales to $BERT_{Large}$ *i.e.,* MPCBert-L$_{w/o\{d\}}$ (best 0.08). (2) baseline benefits from using the pre-trained weights, *i.e.,* MPCBert-B$_{w/o\{d\}}$ (best 0.42) performs better than MPCBert-B$_{w/o\{p, d\}}$ (best 0.23). (3) MPCFormer$_{w/o\{d\}}$ benefits when the base model becomes better, *i.e.,* MPCRoberta-B$_{w/o\{d\}}$ (best 0.62) performs better than MPCBert-B$_{w/o\{d\}}$ (best 0.42).

For MPCFORMER, we decide the number of epochs according to the MSE loss for embedding and Transformer layer distillation, 5 epochs for prediction layer distillation, and batch size 8 for small datasets (CoLA, MRPC, RTE) and 32 for larger ones (MNLI, QQP, SST2, STS-B). We minimize the hyper-parameter tuning for MPCFORMER, since we would like the performance to be an expectation for future researchers using MPCFORMER, who prefer not to tune hyper-parameters. Specifically, we use 5 epochs for MNLI, 5 epochs for QQP, 10 epochs for QNLI, 10 epochs for SST-2, 20 epochs for MRPC, 30 epochs for IMDB 50 epochs for STS-B, 50 epochs for CoLA, 50 epoches for RTE, for the embedding and Transformer layer distillation stage.

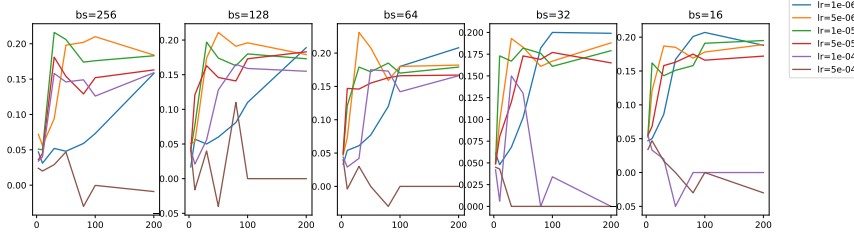

Figure 6: Grid search results for MPCBert-B$_{w/o\{p,d\}}$ on STS-B dataset. X-axis is number of epochs, and Y-axis is correlation (unscaled).

---

[6]We select STS-B because it is a regression task, where performance varies in a large range.

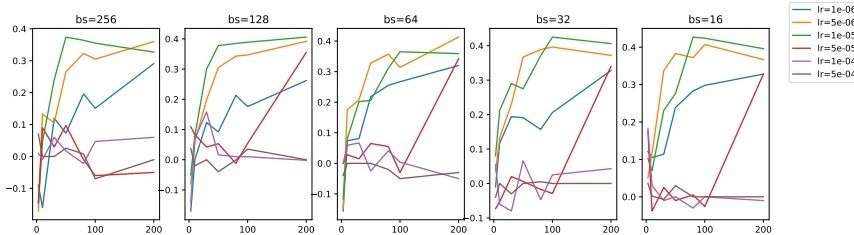

Figure 7: Grid search results for MPCBert-B$_{w/o\{d\}}$ on STS-B dataset. X-axis is number of epochs, and Y-axis is correlation (unscaled).

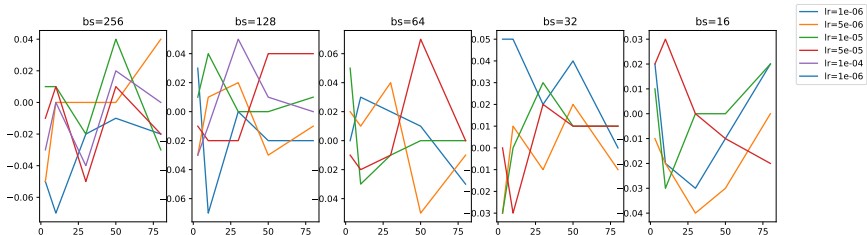

Figure 8: Grid search results for MPCBert-L$_{w/o\{d\}}$ on STS-B dataset. X-axis is number of epochs, and Y-axis is correlation (unscaled).

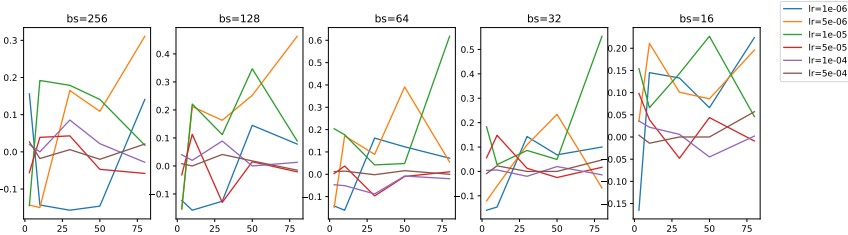

Figure 9: Grid search results for MPCRoberta-B$_{w/o\{d\}}$ on STS-B dataset. X-axis is number of epochs, and Y-axis is correlation (unscaled).

