# OpenReview forum: "MPCFORMER: FAST, PERFORMANT AND PRIVATE TRANSFORMER INFERENCE WITH MPC"
_ICLR.cc/2023/Conference — ICLR 2023 notable top 25%_

### Official Review · Reviewer_BURu · 2022-10-18

**Confidence:** 3
**Correctness:** 4
**Technical Novelty And Significance:** 4
**Empirical Novelty And Significance:** 4
**Recommendation:** 8

**Clarity, Quality, Novelty And Reproducibility:**

The paper is largely written well.

Section 2.2 mentions comparison as an MPC primitive. What is the source of this? My understanding is that comparisons have to be reduced to more primitive operations.

There are a number of space errors: missing space before opening bracket (page 4 twice, pages 5 and 7 once) and unnecessary space before all footnote markers.

I don't think putting quotes around established concepts such as secret sharing and encryption is justified (Figure 1).

The paper should clarify that CrypTen uses two parties plus a helper party because it makes the figures incomparable to pure two-party computation.


**Strength And Weaknesses:**

The paper works out very well how the different parts of their approach contribute to their solution. I'm not aware of these techniques being used in the context of MPC optimization before. I'm positive about the achieved speedups with only a minor loss in accuracy.


**Summary Of The Paper:**

The paper presents a method to transfer a transformer model to a slightly different one that is cheaper to compute in multi-party computation.


**Summary Of The Review:**

Interesting technique applied to multi-party computation

---

> ### Author Response · Authors · 2022-11-10
> **Response to Reviewer BURu**
>
> We thank the reviewer for the positive feedback on our method and presentation. We also thank the detailed review of the paper writing. We summarize questions here and answer them individually:
> - "Primitive" is not precise for the comparison operation: “Section 2.2 mentions comparison … more primitive operations.”
>
> **Response**: We agree with the reviewer. In Crypten ([1]), comparison is implemented by **many more primitive** operations such as AND operations. In the main paper, we do not further describe these implementation details for ease of understanding (they are available in A.1). We have updated the manuscript to use the word **“routine”** instead of “primitive” to avoid confusion.
>
> - Space errors: “There are a number of space errors…”
>
> **Response**: We have updated the spacing issue at Page 4: “Secure Multi-Party Computation (MPC)”, ”Gradient Descent (GD)” and “Knowledge Distillation (KD)”; the spacing issue at Page 5 “Knowledge Distillation (KD)”; the spacing issue at Page 7 “HuggingFace (Wolf et al., 2019)”. We also deleted extra spaces in the footnote in the **updated** manuscript.
>
> - Figure 1 quoting error: “I don’t think …”
>
> **Response**: We have updated Figure 1 and delete the quotation mark around secret-sharing.
> Crypten description is not accurate enough: “The paper should clarify…pure two-party computation.”
> Response: We agree with the reviewer. For ease of illustration, we did not plot the trusted third party(TTP) in Figure 1. We have updated the caption in Figure 1 to specify Crypten uses a TTP to help with the computation.
>
> [1] Knott, Brian, et al. "Crypten: Secure multi-party computation meets machine learning." Advances in Neural Information Processing Systems 34 (2021): 4961-4973.

---

### Official Review · Reviewer_ENWC · 2022-10-25

**Confidence:** 3
**Correctness:** 4
**Technical Novelty And Significance:** 2
**Empirical Novelty And Significance:** 3
**Recommendation:** 8

**Clarity, Quality, Novelty And Reproducibility:**

The idea of using Knowledge distillation to retrain an MPC-friendly surrogate model for the transformer is novel and interesting, and the paper does an excellent job of motivating and presenting it. The authors also provide enough details to reproduce the findings. They also highlight the current work's limitations and offer research directions for future work.

**Strength And Weaknesses:**

Strengths
- Strong empirical performance in terms of accuracy of the final model. For most of the models trained by $\text{MPCFORMER}$, the downstream accuracy is close to the accuracy of the original model. $\textbf{Question:}$ Under this model, would it be possible to have a continual training framework where the model provider could continue the distillation procedure on the partial computations of the teacher model vs. those of the distilled model for randomly chosen client inputs?
- The current models provide a 2x speed up while using a student model of the same size as the teacher. Smaller student models could give even better speedups.
- This is a very general framework that would support various approximations that might be used in the future.

Weakness
- As the activation functions are quadratic, it is possible for the local Lipschitz constants to increase, which would decrease robustness. Current performance measures do not consider other aspects of the model. It would be better to have a more holistic view of model performance for the comparisons.

**Summary Of The Paper:**

The paper provides a procedure for building better surrogate networks for transformer models fine-tuned to a specific downstream task that allows for faster MPC for private sharing with clients while preserving the model's performance (defined by model accuracy). To build a better surrogate, the authors propose a two-stage pipeline, $\text{MPCFORMER}$: 1) Replace bottleneck layers (GeLU, Softmax) in the transformer model with their faster MPC-friendly counterparts (Quad, 2Quad) to form an intermediate model $\mathcal{S}'$ 2) Retrain $\mathcal{S}'$ on the downstream dataset using Knowledge Distillation from the original transformer model. Finally, the authors provide experimental evidence to show the procedure's benefit for various Bert-based transformer models.

**Summary Of The Review:**

The paper addresses an important issue of improving the computation time of MPC for transformer models while preserving the model's performance. To this end, the authors propose a simple and elegant framework that they show to be useful in practice. I think this paper would be of great interest to the ML community.

---

> ### Public Comment · ~Nandan_Kumar_Jha1 · 2022-11-07
> **Idea of using knowledge distillation (KD) in MPC-friendly network**
>
> Dear Reviewer ENWC and Authors,
>
> The idea of using KD  to recover the loss in an MPC-friendly network was first implemented by Jha et al; ICML'21 [1].  In the paper, the teacher model is a baseline network with full ReLUs and the student model is the network with fewer ReLUs.  In fact, it was shown that KD's benefits depend on the neurons' position in the network (Table 1).
>
>
>
>
> [1] Jha et al., DeepReDuce: Relu reduction for fast private inference, ICML'21

---

> > ### Author Response · Authors · 2022-11-09
> > **Reply to Nandan**
> >
> > Hi Nandan,
> >
> > Thanks for referring us to your work. We will cite your paper and discuss it in the related work section in our revised version.
> >
> > First, we want to highlight the differences between our method and yours, and explain why your method cannot easily fit into our Transformer case.
> > * In your case, the alignment is conducted **only in the final logit level**. While MPCFormer conducts distillation in every layer to align the **hidden states, attention scores, the embedding layer and the final logits**. As a result, MPCFormer can support more general MPC-friendly approximations. As shown in our Table 2, training even **without KD** can achieve high performance for certain approximations. However, such success can not be preserved when we introduce more aggressive approximations. **The layer-wise KD in our framework can tolerate all these approximations, which is not shown or mentioned in your paper.** Actually, we believe such robustness can hardly be achieved using solely the logit-level alignment.
> > * Besides, your paper only studied the ReLU activation, which is the major source of slowness in CNNs. However, we find that the activation only takes 18.6% of the total runtime in Transformers, which is much less than softmax that takes 67.8% (Table 2). **This makes the problem for Transformers fundamentally different from the problem for CNNs.** Therefore, it is hard to directly transfer your work to our Transformer case as your approximations for ReLU can only lead to moderate speedup in Transformer. In comparison, we address both sources of slowness in Transformer and propose a new approximation to softmax.
> >
> > Finally and most importantly, we want to highlight that our contribution is the **framework**, not just “introducing KD in MPC-friendly networks”. Our overall goal is to build a simple and general framework to support **various transformer weights and approximations, so that future model providers can easily adopt the framework and can flexibly choose their preferred approximations**. Instead, the layer-wise KD is just one technique we use to achieve our goal, which can be replaced with any advanced technique in the future.

---

> > > ### Public Comment · ~Nandan_Kumar_Jha1 · 2022-11-09
> > > **Logit-based distillation vs layer-wise distillation**
> > >
> > > Dear Authors,
> > >
> > > Thanks for your reply!
> > >
> > > I agree and it is already shown that distilling from multiple layers performs better than logit-based KD (**not always**) if distillation is performed in an intelligent manner (for example, Table 1 and Table 2 in [1]). I believe that a comparison between logit-based KD (popularly known as Hinton's KD) and the distillation method proposed in your paper would highlight the importance of distilling from different stages in the transformer (when softmax and GELUs are approximated).
> > >
> > > Thanks again for your detailed reply!
> > >
> > >
> > > [1] Chen et al, Distilling Knowledge via Knowledge Review, CVPR'21

---

> > > > ### Author Response · Authors · 2022-11-11
> > > > **Reply to Nandan**
> > > >
> > > > Hi Nandan,
> > > >
> > > > Thanks for the reference. We ran experiments on several datasets using only logits-level distillation with ”Quad”+”2Quad”. **This shows logit-level distillation is not sufficient for Transformer models**.
> > > >
> > > >
> > > > | | MNLI-m | MNLI-mm | CoLA | RTE |
> > > > | ----- | ----- | ----- |  ----- |  ----- |
> > > > | Original Transformer | 84.7 | 85.0 | 57.8 | 69.7 |
> > > > | MPCFormer | **84.9** | **85.1** | **52.6** | **64.9** |
> > > > | Logit-level | 33.0 | 35.2 | 0.0 | 47.3 |

---

> > > > > ### Public Comment · ~Nandan_Kumar_Jha1 · 2022-11-12
> > > > > **Thanks for the data**
> > > > >
> > > > > Dear Authors,
> > > > >
> > > > > Thanks for the data, looking forward to seeing the paper in ICLR!

---

> ### Author Response · Authors · 2022-11-10
> **Response to Reviewer ENWC**
>
> We thank the reviewer for the positive feedback on the importance of the problem, the effectiveness, simplicity, and generality of our framework, and the comment that “this paper would be of great interest to the ML community.” We summarize the concerns and answer them individually below:
>
> - Continual training: “Under this model…randomly chosen client inputs?”
>
> **Response**: We agree with the reviewer that the continual learning setting is of great importance and interest. It would  be great, however, if you could **clarify** the sentence “on the partial computations of the teacher model vs. those of the distilled model for randomly chosen client inputs”. In our understanding, this scenario is that the model provider has new data, e.g., from other data streams, and wants to improve the quality of the distilled model. An immediate  idea is that, in our setting, the model provider can first use the new data to **further fine-tune** the teacher model, and then continue the knowledge distillation stage on the **previously distilled student model with the newly fine-tuned teacher model**. We plan to investigate how to do continual learning in MPCFormer in possibly a more efficient way, e.g., how many training iterations are enough for a certain amount of new data, in an exciting future direction.
>
> - Robustness: “As the activation functions are quadratic … for the comparisons.”
>
> **Response**: We thank the reviewer for raising the interesting question on the robustness after approximation. We applied some empirical **text adversarial attacks** to evaluate the adversarial robustness of the $\text{BERT}_\text{BASE}$ backbone before and after approximations [1]. As shown in Table below, our student model has a slight increase  (**<0.05**) in terms of attack success rate (ASR) over the three score-based attacks. But the student model has a lower ASR with the gradient-based attack HotFlip. Overall, we do not see significant changes in robustness.
>
> Table 1. Sanity accuracy (SA) and attack success rate (ASR) against various text attacks on $\text{BERT}_\text{BASE}$ in SST-2 dataset. The TextFooler, PWWS, and BERT-ATTACK are score-based attacks, and HotFlip is a gradient-based attack. For ASR, lower is better.
>
> |                                                  | SA    | TextFooler | PWWS | BERT-Attack | HotFlip | Speedup |
> | -----------                                    | ------- |-----------     |----------- |-----------         |---------  |-----------    |
> | Original Transformer (teacher) | **0.93**  | **0.76**          | **0.78**      | **0.88**               | 0.55     | 1.0x         |
> | MPCFormer (student)              | 0.92  | 0.78           | 0.82     | 0.91               | **0.51**     | **2.2x**         |

---

### Official Review · Reviewer_tn3E · 2022-10-26

**Confidence:** 3
**Correctness:** 4
**Technical Novelty And Significance:** 2
**Empirical Novelty And Significance:** 4
**Recommendation:** 8

**Clarity, Quality, Novelty And Reproducibility:**

**Clarity.** The writing, figures, and tables are exceptionally clear.

**Quality.** Good. The paper appears to be technically sound and evaluations are thorough.

**Novelty.** This is a systems paper. Algorithmic and theoretical contributions are limited.

**Reproducibility.** Key details are comprehensively described such that competent researchers will be able to easily reproduce the main results.

**Strength And Weaknesses:**

# Strengths

* The proposed approach is highly practical, as it can be easily adapted to fit a range of transformer architectures and tasks.

* The paper is the first to tackle private transformer inference with MPC.

* A novel MPC-friendly Softmax approximation, dubbed "2Quad" is proposed, which provides a significant speed advantage compared to the existing "2Relu" Softmax approximation.

* Through evaluation and strong results. Experimental evaluations across a range of language tasks and a range of transformer architectures validate the efficacy of the proposed approach relative to existing methods, both in terms of speed and performance. Ablation study demonstrates the importance of the proposed knowledge distillation step, model initialization protocol, and Softmax approximation.

# Weaknesses

* Limited algorithmic and theoretical contributions.



**Summary Of The Paper:**

The paper proposes MPCFormer, a novel two-stage approach for private transformer inference with MPC. The first stage replaces bottleneck functions in the pre-trained transformer (ie, functions with high computational/communication complexity under MPC), with MPC-friendly approximations. The second stage uses knowledge distillation to map the outputs of multiple layers of the MPC-friendly model to those of the original transformer. Additionally, a novel MPC-friendly Softmax approximation, dubbed "2Quad" is proposed, which provides a significant speed advantage compared to the existing "2Relu" Softmax approximation. Experimental evaluations on a range of language tasks with a range of transformer architectures validate the efficacy of the proposed approach relative to existing methods, both in terms of speed and performance.

**Summary Of The Review:**

Strong systems paper with limited algorithmic/theoretical novelty.

---

> ### Author Response · Authors · 2022-11-10
> **Response to Reviewer tn3E**
>
> We thank the reviewer for the positive feedback on the practicality and presentation, and it is encouraging to hear our paper is  a “strong system paper”. Indeed, we primarily focus on studying the system-level pattern, and designing the overall framework with good usability and effectiveness. However, we believe that for the private Transformer inference problem, a solution where the model provider can easily **adopt** is of great importance. MPCFormer is such a solution that allows the model provider to easily convert a Transformer model into one that is better suited in the private inference service. Comprehensive evaluations have demonstrated the usability and effectiveness of MPCFormer, and thus we believe MPCFormer makes substantial contributions.

---

### Official Review · Reviewer_dCKB · 2022-11-02

**Confidence:** 2
**Correctness:** 3
**Technical Novelty And Significance:** 3
**Empirical Novelty And Significance:** 3
**Recommendation:** 6

**Clarity, Quality, Novelty And Reproducibility:**

Some of the technique details are not clearly, for example, in Figure 1, what is the communication volume of each step and the computation load on the two sides. In Figure 2, the run-time breakdown is interweaving, clearly the excution is not proportional to the FLOP operation of each step in the original transformer computation, where at least more than 90\% of computation is for MatMul, why does the breakdown look so different under MPC? What is the communication paradigm and overhead introduced by MPC? Without these details, it is hard to understand this results.

There is a lack of theoretical analysis of the approximating GeLU and softmax, essentially, is there any guarantee for these relaxations? Or is this just based on some empirical observation?

The source code is attached to provide good Reproducibility.

**Strength And Weaknesses:**

Strength:

+ This paper proposes an interesting question for privacy-preserve inference of transformer models.


Weaknesses:

- This is a lack of theoretical analysis on the relaxation of the computation.

- The performance breakdown is unclear without concrete Flops analysis under the original computation of transformers and the computation of the MPC setting.


**Summary Of The Paper:**

This paper proposed MPCFORMER to enable fast inference under the privacy constraint of mutlit-party computation for transformer based models. Some speed-up is achieved as a result of the relaxation of the proposed method in the experiments.

**Summary Of The Review:**

This paper proposes an interesting problem about privacy preserving inference of transformer based models, however, the technique section is not clearly stated and there is a lack of theoretical analysis of the approximation of the computation.

---

> ### Author Response · Authors · 2022-11-10
> **Response to Reviewer dCKB (1/2)**
>
> We thank the reviewer for the positive feedback on “This paper proposes an interesting problem”. We summarize individual concerns and answer them below:
> - Technical section is not clearly stated: (1) What is the communication volume and computation load on both sides in Figure 1? (2) Why is the breakdown not proportional to FLOP in MPC? (3) What are the communication paradigm and overhead introduced by MPC?
>
> **Response**: In fact, we have attempted to answer these questions in subsection 2.2 both qualitatively and quantitatively. (1) Qualitatively, we have introduced the secret-sharing scheme in MPC, and how deep learning operators, e.g., multiplications, and Softmax are **implemented** with the secret-sharing. Please refer to subsection 2.2 (4th paragraph), i.e., “it requires one extra round of communication … Any computational operations composed by these routines would result in extra complexity in MPC.” for more details. In short, **The major source of slowness is the communication instead of computation in MPC, and thus FLOP is no longer an accurate estimation metric for the running time**. (2) Quantitatively, we profiled the communication statistics in Table 1. For instance, the whole BERT_BASE inference takes 59.0 seconds, with **communication** in Softmax functions **alone** taking 34.1 seconds. The heavy communication overhead is also observed in previous studies ([1], [2]). Additionally, we show the computation and communication statistics for different MPC-friendly approximations for GeLU and Softmax in Figure 4.
>
> We answer other specific questions from the reviewer.
>
> **Q1**: “in Figure 1, what is the communication volume of each step and the computation load on the two sides?”
>
> **A1**: Overall, computation takes **21%** of the overall time, and communication takes **79%** of the overall time for the setting in Figure 2. More precisely, communication takes 34.1, 9.6, and 2.5 seconds for Softmax, GeLU, and MatMul, respectively. Computation takes 5.9, 1.4, and 5.0 seconds for them. The MPC system we use in the paper uses **All-reduce** to implement intermediate communication, where both parties have the same communication load. And they have similar computation loads (shown in the multiplication example in 2.2, where both parties are computing the same function locally with their own secret shares). Thus, **the time breakdown is similar for both parties**.
>
> **Q2**: “Why is the breakdown not proportional to FLOP in MPC?”
>
> **A2**: **Since communication dominates the execution, FLOP is no longer an accurate estimation for the running time**. For instance, MatMul should take at least 90% of the execution time in plain-text Transformer inference, but its communication overhead is low (2.5 seconds versus 34.1 for Softmax). Thus, MatMul only takes 7.4 seconds in total, which is 12.7% of the overall running time.
>
> **Q3**:  “What is the communication paradigm and overhead introduced by MPC?”
>
> **A3**:  The paradigm is that parties execute the **same** code to jointly compute functions. When they encounter functions that incur **synchronization** (e.g. multiplication [3]), they communicate necessary intermediate values to proceed (e.g. using All-Reduce). In the MPC system we use, both parties hold local secret shares and use one round of All-Reduce when executing a multiplication. More details on the communication message are provided in Section 2.2. This communication overhead is **unavoidable** when evaluating many functions in MPC though different systems may implement functions differently.
>
> We hope these additional explanations can address your questions above. **We have updated the manuscript for better clarity (in section 2 and section A.1)**.

---

> ### Author Response · Authors · 2022-11-10
> **Response to Reviewer dCKB (2/2)**
>
> - Lack of theoretical analysis of the approximated GeLU and Softmax. “There is a lack of theoretical analysis of the approximating GeLU and softmax, essentially, is there any guarantee for these relaxations? Or is this just based on some empirical observation?”
>
> **Response**: we interpret the word “theoretical analysis” in three aspects: speed, privacy, and accuracy.
>
> - **For speed**, we analyze communication round and volume quantitatively in Section 2. This analysis explains why certain approximations are faster than the original operations. For instance, each comparison routine requires 7 rounds of communication while each multiplication routine only requires 1 round. This study motivates our proposed ``2Quad” approximation, where we replace ReLU in "2ReLU" approximation by quadratics. This speed up is also verified by our empirical studies (Figure 4).
>
> - **For privacy**, the privacy of approximated GeLU and Softmax **is guaranteed** by MPC ([4]) because MPC keeps the model and data private throughout the inference process; we do not modify the underlying MPC engine.
>
> - **For accuracy**, it is hard to predict or provide guarantees for deep learning performance, especially for large-scale Transformer models. Instead, we provide empirical evaluation across different Transformer weights, approximations, and various datasets, and show that the knowledge distillation stage in MPCFormer works well with these approximations in practice.
>
> [1] Mishra, Pratyush, et al. "Delphi: A cryptographic inference service for neural networks." 29th USENIX Security Symposium (USENIX Security 20). 2020.
>
> [2] Juvekar, Chiraag, Vinod Vaikuntanathan, and Anantha Chandrakasan. "{GAZELLE}: A low latency framework for secure neural network inference." 27th USENIX Security Symposium (USENIX Security 18). 2018.
>
> [3] Beaver, Donald. "Efficient multiparty protocols using circuit randomization." Annual International Cryptology Conference. Springer, Berlin, Heidelberg, 1991.
>
> [4] Knott, Brian, et al. "Crypten: Secure multi-party computation meets machine learning." Advances in Neural Information Processing Systems 34 (2021): 4961-4973.

---

> ### Author Response · Authors · 2022-11-29
> **We are more than happy to know your further comments**
>
> Dear Reviewer dCKB,
>
> Once again, we would like to express our sincere gratitude for your review. We hope that our response has adequately addressed your concerns. If you have any additional feedback concerning our current draft, we would be delighted to hear it and address your points.
>
> Best,
> The authors

---

### Author Response · Authors · 2022-11-10
**General Response**

We thank the reviewers for their positive feedback. We are encouraged that they appreciate our paper for the following reasons:
- The problem of fast, accurate, and private Transformer inference is of interest and importance.
- The approach is well-motivate.
- The framework is simple, general, and practical.
- Evaluations are thorough and show strong effectiveness.
- The paper is largely well-written.

We summarize some of the main changes to the paper below.
- Description of the communication pattern in the background section.
- A robustness experiment demonstrating a more holistic effect of our framework.
- wording and formatting such as extra spaces.

**As the revised manuscript is constrained by the page limit, we temporarily put (ii) and part of (i) in the appendix and highlighted them with blue color**. We will merge them with the main text in the final version. We will respond to each reviewer individually.

---

### Decision · Program_Chairs · 2023-01-20

**Decision:**

Accept: notable-top-25%

**Justification For Why Not Higher Score:**

As commented by the multiple reviewers, the property of the introduced approximation is not thoroughly analysed theoretically. The introduced MPC-friendly approximation has moderate novelty.

**Justification For Why Not Lower Score:**

The proposed method for private inference under MPC has a general and practical value. Experiments show good accuracy and speed improvement compared to the original model and prior works.

**Metareview: Summary, Strengths And Weaknesses:**

This paper studies how to speed up private Transformer inference under the multi-party computation (MPC) setting. It proposes a two-stage approach, replaces bottleneck functions in the original model with MPC-friendly approximations in the first stage, and run knowledge distillation to improve the approximation to the orignal network in the second stage. It introduces "2Quad" operator to approximate the communication-expensive softmax operator.

Strengths:
- It's a simple and practical method that can be applied to various Transformer models and easy to be adopted.
- Experiments on multiple Transformer architectures and benchmarks show good speed up with only a slight drop in accuracy.
- Authors provide additional experiments to compare the proposed layer-wise knowledge distillation with logit-based KD in prior work during reviewing discussion. It would be nice to include the results in the final revision.

Weaknesses:
- Lack of theoretical analysis on the approximation or save of computation time. As commented by a reviewer, this paper is more of a system paper and it focuses on the system-level pattern. Nonetheless, quantifying the effect of approximation error for large-scale transformer model is difficult given existing theoretical analysis tools. Therefore, this weakness should not be held as a critical weakness against acceptance.


**Note From Pc:**

if the above contains the word "oral" or "spotlight" please see: "oral" presentation means -> notable-top-5% and "spotlight" means -> notable-top-25%. As stated in our emails, we are disassociating presentation type from AC recommendations

**Summary Of Ac-Reviewer Meeting:**

N/A